# FEATURE SELECTION IN THE CONTRASTIVE ANALYSIS SETTING

## ABSTRACT

The goal of unsupervised feature selection is to select a small number of informative features for use in unknown downstream tasks. Here the definition of "informative" is subjective and dependent on the specifics of a given problem domain. In the contrastive analysis (CA) setting, machine learning practitioners are specifically interested in discovering patterns that are enriched in a *target* dataset as compared to a *background* dataset generated from sources of variation irrelevant to the task at hand. For example, a biomedical data analyst may wish to find a small set of genes to use as a proxy for variations in genomic data only present among patients with a given disease as opposed to healthy control subjects. However, as of yet the problem of feature selection in the CA setting has received little attention from the machine learning community. In this work we present CFS (Contrastive Feature Selection), a method for performing feature selection in the CA setting. We experiment with multiple variations of our method on a semi-synthetic dataset and four real-world biomedical datasets, and we find that it consistently outperforms previous state-of-the-art methods designed for standard unsupervised feature selection scenarios.

## 1 INTRODUCTION

In many scientific domains, it is increasingly common to see high dimensional datasets consisting of observations with many features. However, not all features are created equal: oftentimes, noisy information-poor features may obscure the natural structures underlying a dataset and impair the performance of machine learning algorithms on downstream tasks. As such, it is often desired to select a small number of informative features and consider only those features when performing future analyses. For example, biologists may seek to measure the expression levels of only a small number of informative genes and then use these measurements in a variety of future prediction tasks, such as disease subtype prediction or cell type classification. Measuring only a subset of features may also yield a number of other benefits, such as reduced experimental costs, enhanced interpretability, and better generalization of models trained to perform downstream tasks.

While feature selection is relatively straightforward when prior knowledge on the structure of the data is available (e.g. class labels in a supervised setting), such information is often not available *a priori* and thus unsupervised methods must be used. In the unsupervised setting, feature selection algorithms seek to select features that are useful for arbitrary downstream tasks. A diverse array of prior work exists on unsupervised feature selection, with different methods taking different approaches to quantify feature quality. For example, some methods are designed to select features that provide good clustering results (Mitra et al., 2002; Lu et al., 2007), while others are designed to pick features that preserve local structures in the data (Dy & Brodley, 2004; Boutsidis et al., 2009; Cai et al., 2010).

In many domains, data analysts are specifically interested in patterns that are enriched in one dataset, referred to as the *target*, as compared to a second related dataset, referred to as the *background*. Such target and background dataset pairs arise naturally in many settings; data from healthy controls versus a diseased population, from pre-intervention and post-intervention groups, or signal-free versus signal-containing recordings all form natural background and target pairs (Abid et al., 2018). In each of these scenarios, the target dataset likely shares some set of uninteresting nuisance variations with the background dataset, such as population level variations or sensor noise.

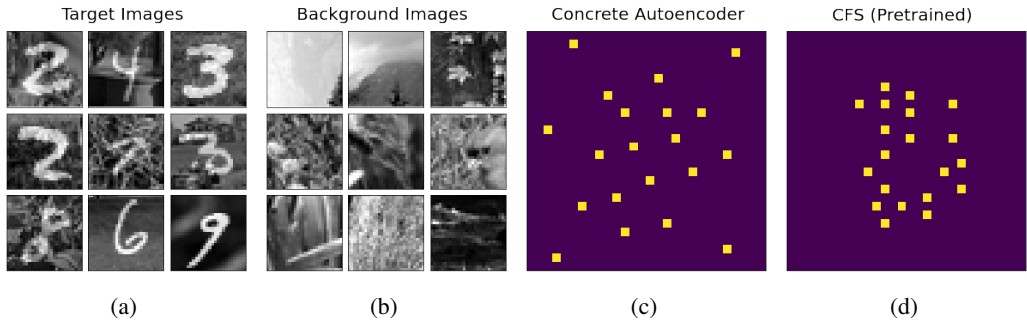

| Target Images | Background Images | Concrete Autoencoder | CFS (Pretrained) |
| --- | --- | --- | --- |
| (a) | (b) | (c) | (d) |

Figure 1: **Visualizing CFS applied to the Grassy MNIST dataset to select** $k = 20$ **features.** (**a**): This semi-synthetic target dataset was constructed by superimposing handwritten digits from the MNIST dataset onto images of grass from ImageNet. Our goal here is to select features that allow us to classify images by digit. (**b**): For our background dataset, we used a separate set of images of grass. (**c, d**): Features selected (yellow) by a concrete autoencoder (**c**), a state of the art feature selection method, and by a variant of our method CFS (**d**). Features selected by CFS are concentrated towards the center of the images, where our superimposed digits lie. On the other hand, features selected by the concrete autoencoder are more diffuse throughout the image.

Unfortunately, when these nuisance variations comprise the majority of the overall variance in a dataset, standard feature selection methods may not select the features that best reflect the variations of interest. For example, suppose we wanted to collect gene expression measurements from cancer patients for only a subset of genes to better understand cancer-related differences between patients, such as differences between molecular subtypes of cancer. Understanding such differences is of great importance to the cancer research community, as intra-cancer variations are closely related to disease progression and treatment response (Stanta & Bonin, 2018). With knowledge on different varieties of cancer rapidly evolving and new treatments consistently being released at a fast pace (Kinch, 2014), it is unlikely that we would have access to a single set of informative labels that would enable us to use supervised techniques for this task. Unfortunately, selecting features using fully unsupervised methods is also not straightforward, as the variations in this data can be broken down into two components: those shared across healthy and cancerous tissue (e.g. variations due to demographic factors), and those only present in cancerous tissue. If the cancer-specific variations are subtle compared to the shared variations, standard unsupervised feature selection algorithms may not select a suitable set of genes for our analyses.

Isolating these salient variations present only in a target dataset is the subject of *contrastive analysis* (CA; Zou et al. (2013)), and a number of algorithms have been proposed that extend unsupervised learning methods to the CA setting. For example, Abid et al. (2018) extended principal component analysis to the CA setting, and many recent works (Li et al., 2020; Severson et al., 2019; Abid & Zou, 2019; Ruiz et al., 2019; Jones et al., 2021) have focused on developing methods for learning contrastive latent variable models. However, feature selection in the CA setting has not been studied in depth by previous work.

The contributions of this paper are the following:

1. *We introduce the problem of feature selection in the CA setting (Section 2).* We do so by connecting feature selection to the specific data generating process assumed in CA.

2. *We propose CFS (Contrastive Feature Selection), a method for feature selection that specifically selects features useful for downstream CA tasks (Section 4).* At a high level, CFS aims to select a set of features that best reflect variations enriched in a target dataset but which are not present in the background dataset.

3. *We validate our approach through extensive experiments (Section 5).* We begin by applying CFS to a semi-synthetic dataset to better understand when it may succeed as well as potential failure cases. We then apply CFS to four real-world biomedical datasets and find that it consistently outperforms standard fully unsupervised feature selection algorithms.

**Target**   **Background**

Figure 2: **Data generating processes for target and background samples.** Background latent variables $z$ are common to the two processes. A second set of salient latent variables $s$ are used to generate target points, while these variables are fixed for background points. The salient variables (and *not* the background variables) are used to generate labels $y$ for target points. Observed values are shaded, and square nodes represent constant values.

## 2 PROBLEM FORMULATION

We now describe the problem of feature selection in the CA setting. Recall that the goal of CA is to isolate variations only present in a target dataset from those present in both the target and background datasets. To capture this idea, we assume our data can be modeled according to the following process: let $X = \{x_i\}_{i=1}^{n} \in \mathbb{R}^{n \times d}$ denote our target dataset consisting of samples of interest. We assume that our $x_i$ are drawn i.i.d. from a random process that depends on two sets of latent variables $z_i$ and $s_i$. We use $s$ to refer to variables that are *salient* in our analysis, while $z$ denotes *background* variables that are not relevant to us. Our observed $x_i$ are then drawn from a function of our two sets of latent variables, i.e.:

$$x_i \sim p(x|z_i, s_i).$$

We also assume that each data point $x_i$ is associated with labels $y_i$ drawn from another function that depends on $s$, i.e.:

$$y_i \sim q(y|s_i).$$

That is, our labels depend *solely* on the salient latent variables, and not on the background variables. Our goal is to select some subset of features $S \subseteq \{1, 2, \ldots, d\}$ of size $|S| = k$ that can be used to infer $y$. Were our $y$ known in advance, selecting appropriate features would be straightforward using standard supervised feature selection techniques. However, we assume that $y$ is not given *a priori*, and so we must instead select a feature set $S$ suitable for arbitrary downstream tasks that depend on the salient latent variables $s$.

When run on a target dataset alone, standard unsupervised feature selection algorithms will not necessarily select features that contain much information about $s$ (or, by extension, future labels $y$). To isolate these salient variations, we can leverage a background dataset $B = \{b_j\}_{j=1}^{m} \in \mathbb{R}^{m \times d}$ for which we assume that samples are generated only from the irrelevant latent factors with the salient variables fixed to some constant $s'$. We note that both our target and background samples are assumed to be drawn from the same data generating process $p(\cdot)$. Moreover, we do not assume that we have the same number of samples in the target and background datasets (i.e., $n \neq m$), and we do not assume that there is any special relationship between samples $i$ and $j$ from the target and background datasets, respectively. We depict this data generating process in Figure 2.

Leveraging such a background dataset amounts to a form of weak supervision that arises naturally in many real-world settings. For example, suppose we wanted to analyze data from patients receiving an experimental treatment in a clinical trial to better understand differences in how the treatment impacts the patients who receive it. In this case, a suitable background dataset would consist of data from control patients drawn from a similar population but administered a placebo instead of a real

treatment, and our salient latent factors of interest would correspond to variations present *only* in patients that received the real treatment.

## 3 RELATED WORK

**Contrastive Analysis.** The idea of CA has previously been used to extend a number of unsupervised learning methods. Zou et al. (2013) initially adapted mixture models for CA. Recent work (Abid et al., 2018) adapted principal component analysis to the contrastive setting so that a set of salient principal components are learned that summarize variations of interest in a target dataset while ignoring uninteresting variations. Other recent works (Li et al., 2020; Abid & Zou, 2019; Severson et al., 2019; Ruiz et al., 2019; Jones et al., 2021) have since developed probabilistic latent variable models for CA. As part of their work, Severson et al. (2019) experimented with adding a sparsity prior to one of their proposed contrastive latent variable models to perform feature selection. However, this penalty was designed for use only in fully supervised scenarios; the penalty was manually tuned by observing how well pre-labelled classes of target points separated in the resulting latent space, and this procedure is thus not suitable for use in the scenario described in Section 2.

We note that CA can be seen as a specific case of the problem of learning disentangled representations (Bengio, 2013), and which is the subject of much previous work (Higgins et al., 2017; Kim & Mnih, 2018; Chen et al., 2018; Mathieu et al., 2019; Lopez et al., 2018; Kumar et al., 2018; Burgess et al., 2018; Locatello et al., 2019). However, unlike the general case, in which each feature encodes a semantically meaningful attribute and is invariant to the other features, in CA we only require that background and salient features are disentangled; moveover, in the CA setting we are able to leverage a background dataset as a form of weak supervision not available in the general case, thus making our problem more tractable.

Finally, we note that CA is unrelated to the similarly named field of contrastive representation learning (Chen et al., 2020a; Schroff et al., 2015; Chen et al., 2020b; Chopra et al., 2005).

**Feature Selection.** Feature selection has been extensively studied in the machine learning literature. Much of this work has been devoted to selecting features in the supervised setting, in which each sample is associated with a label that can be used to guide the selection process. Such methods can be divided broadly into three categories. Filter methods (Battiti, 1994; Peng et al., 2005; Estévez et al., 2009; Song et al., 2007; 2012; Chen et al., 2017) remove irrelevant features before model training by computing per-feature relevance scores based on some statistical measure. Wrapper methods (Kohavi & John, 1997; Stein et al., 2005; Zhu et al., 2007; Reunanen, 2003; Allen, 2013) assess the quality of subsets of features by training new models for each subset. Finally, embedded methods (Tibshirani, 1996; Yamada et al., 2020; Li et al., 2016; Scardapane et al., 2017; Feng & Simon, 2017) jointly learn a model while selecting an optimal subset of features.

In this work we focus on feature selection in settings where we do not have access to per-sample labels. In such scenarios we must select a set of features that are useful for arbitrary downstream tasks, and a variety of criteria have been proposed to perform fully unsupervised feature selection, including: selecting clusters of similar features (Mitra et al., 2002; Lu et al., 2007), using spectral information to find features that best preserve the local structure of the data (He et al., 2005; Lindenbaum et al., 2020), finding features that preserve local discriminative information (Yang et al., 2011), and selecting features that best preserve local clustering structure (Dy & Brodley, 2004; Boutsidis et al., 2009; Cai et al., 2010).

Despite this diversity of approaches, to our knowledge no previous works have addressed feature selection for CA as formulated in Section 2 and taken advantage of the weak supervision available in this setting.

## 4 METHOD

We aim to select $k$ features that are informative of the salient latent variables $s$ (and thus, any potential future labels $y$) in our data generating process from Section 2. Such a procedure must be performed delicately, as standard unsupervised feature selection methods may select features that capture much of the overall variance in a dataset but not the (potentially subtle) variations due to $s$.

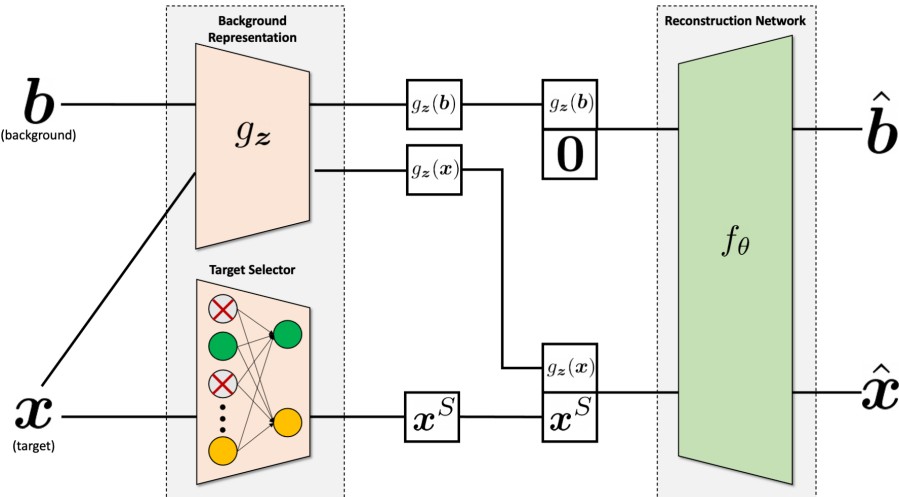

Figure 3: **CFS model architecture.** Here we illustrate the general architecture of CFS-based methods for feature selection. We use our background dataset to learn an embedding function $g_{\boldsymbol{z}}$ that only captures variations due to the latent variables $\boldsymbol{z}$ that are present in both the background and target datasets. We also train a concrete selector layer to select features that best reflect the salient latent variables used to generate target points. This selector layer is trained along with a reconstruction function $f_\theta$ that is trained to reconstruct target data points using the concatenation of the output of $g_{\boldsymbol{z}}$ and the features selected by our gates, and to reconstruct background data points using the concatenation of the output of $g_{\boldsymbol{z}}$ and a zero vector. Here $\boldsymbol{x}^S$ refers to a target sample restricted to the set of features $S$.

To tackle our problem, we thus need a way to isolate the $k$ features that best capture the variations in our dataset specifically due to $\boldsymbol{s}$. To do so, we will use the following intuitive idea: suppose we have some function $g_{\boldsymbol{z}} \colon \mathbb{R}^d \to \mathbb{R}^{k'}$ with $k' < d$ that takes in one of our target data points $\boldsymbol{x}_i$ as input and returns a low dimensional representation of $\boldsymbol{x}_i$ that captures variations due to $\boldsymbol{z}_i$ but does not contain information on $\boldsymbol{s}_i$. We call this function a *background representation* function. Assuming we have such a $g_{\boldsymbol{z}}$, we then select a set of features $S$ that minimize the error of a function $f_\theta \colon \mathbb{R}^{k+k'} \to \mathbb{R}^d$ parameterized by $\theta$ that is trained to use our background representations as well as our selected features to reconstruct the original target data points. That is, we attempt to optimize

$$\underset{S,\theta}{\arg\min}\, \mathbb{E}||f_\theta(g_{\boldsymbol{z}}(\boldsymbol{x}_i), \boldsymbol{x}_i^S) - \boldsymbol{x}_i||_2^2,$$

where $\boldsymbol{x}_i^S$ denotes the target data point $\boldsymbol{x}_i$ restricted to the set of features $S$. In other words, we select the set of $k$ features $S$ that capture the variations present in our target dataset but which are missed by $g_{\boldsymbol{z}}$. We call our approach CFS (Contrastive Feature Selection).

To implement CFS, we must make two choices. First, assuming we already have some $g_{\boldsymbol{z}}$, we need a method for selecting our set of features $S$. To do so, we leverage previously proposed stochastic gating layers (Yamada et al., 2020; Balın et al., 2019; Lindenbaum et al., 2020) developed as part of recent state of the art feature selection methods. Such methods select features by optimizing continuous approximations of discrete variables at the input layer of a neural network; selected variables are multiplied with a 1 and allowed to pass through the gates, while rejected features are multiplied with a 0. In our experiments we train a concrete selector layer (Balın et al., 2019) to select features that, when concatenated with the low-dimensional representation from $g_{\boldsymbol{z}}$ and fed to $f_\theta$, allow us to reconstruct target data points faithfully; this idea is depicted in Figure 3, and we refer the reader to Appendix A for details on the concrete selector layer.

Next, we need a procedure for learning $g_{\boldsymbol{z}}$. Learning such a function is not straightforward, as it requires $g_{\boldsymbol{z}}$ to capture *only* variations due to $\boldsymbol{z}$, and not those due to $\boldsymbol{s}$; if information leaks from $\boldsymbol{s}$

into $g_z$, we are less likely to select the features that provide the most information on $s$. However, with the help of a background dataset assumed to be generated only from $z$, we may be able to successfully isolate these variations. In this work we explore multiple potential approaches for learning $g_z$:

1. *Joint*: We let $g_z$ take the form of a multi-layer perceptron, and train it jointly with our concrete selector layer and reconstruction function $f_\theta$ using both our target and background datasets. When reconstructing a target data point using $f_\theta$, we concatenate the output of $g_z$ with the features selected by our concrete selector layer. On the other hand, when reconstructing a background data point, we concatenate the output of $g_z$ with a zero vector meant to represent the absence of salient variations. Such an approach is highly expressive and likely to lead to good reconstructions of our data. However, without additional constraints on $g_z$ it is also possible that information will leak between $g_z$ and our concrete selector layer, leading our model to select lower-quality features. To test this hypothesis, we also experiment with two other approaches for learning $g_z$ designed to reduce the possibility of information leakage.

2. *Pretraining*: We first train a standard autoencoder solely on our background dataset. We then use our autoencoder's encoder network as $g_z$, freeze its weights, and then learn $f_\theta$ and our concrete gates layer as described for the Joint model. By fixing the weights of $g_z$, we hope to prevent it from encoding salient variations when applied to target data points.

3. *Gates*: We let $g_z$ take the form of a second concrete selector layer and otherwise train it using the same approach as our Joint model. By constraining $g_z$ to perform feature selection, as opposed to general representation learning, we hope to reduce the possibility for information to leak between $g_z$ and our target-point-specific concrete selector layer.

## 5 Experiments

Here we validate our CFS approach by applying it to a number of datasets. We begin by reporting results from a suite of experiments on a semi-synthetic Grassy MNIST dataset. We then report results obtained from four publicly available collections of biomedical data. For each of these collections, points in the target dataset have ground-truth class labels. However, distinguishing between these classes may be difficult using features selected by algorithms not designed for the CA setting. In our experiments we compared our proposed training schemes against a set of fully unsupervised feature selection methods to illustrate the benefits of our weakly supervised approach. These baseline methods include: the Laplacian score (He et al., 2005), MCFS (Cai et al., 2010), PFA (Lu et al., 2007), the concrete autoencoder (CAE; Balın et al. (2019)), and the recently proposed gated Laplacian score (Lindenbaum et al., 2020). We refer the reader to Appendix B for details on dataset preprocessing and Appendix C for details on model implementation and training. For all experiments, we divide our data using an 80-20 train-test split. Our code for downloading and preprocessing these datasets as well implementations of our models are available at www.placeholder.com.[1]

### 5.1 Grassy MNIST

The Grassy MNIST dataset, originally introduced in Abid et al. (2018), is constructed by superimposing pictures with the "grass" label from ImageNet (Deng et al., 2009) onto handwritten digits from the MNIST dataset (LeCun et al., 2010). For each MNIST digit, a randomly chosen grass image was cropped, resized to be 28 by 28 pixels, and scaled to have double the amplitude of the MNIST digit before being superimposed onto the digit. Our goal here is to classify images based on their digit class. For a background dataset, we used a separate set of images consisting only of grass. We note that, although Grassy MNIST is an image dataset, the MNIST digits in each image are centered and thus we can treat each pixel as a separate feature as has been done with the standard MNIST dataset in previous works on feature selection (Balın et al., 2019; Yamada et al., 2020; Lindenbaum et al., 2020).

---

[1]Code available in supplementary materials and will be made public upon acceptance.

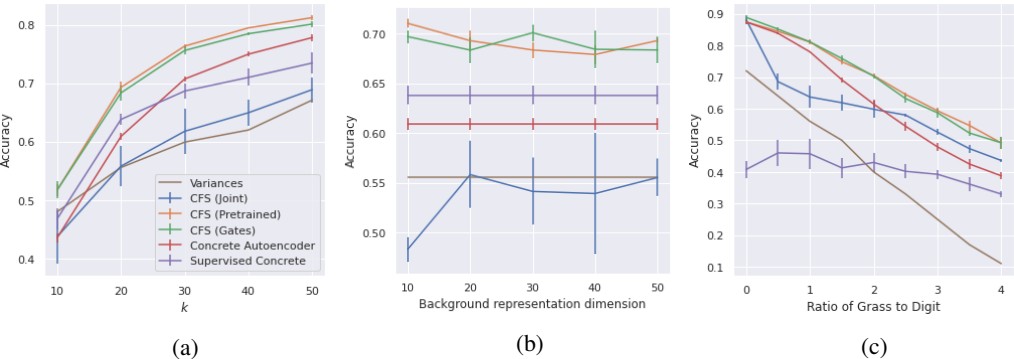

(a)               (b)               (c)

Figure 4: **Downstream classification accuracies and sensitivity analyses of CFS on Grassy MNIST: (a)**: We quantitatively assess the quality of each method's selected features by training an extremely randomized trees classifier (Geurts et al., 2006) to classify Grassy MNIST images by their digit class using varying numbers of selected features $k$. We report mean accuracies over five initializations, and error bars correspond to standard deviation. For all tested values of $k$, we find that our Pretrained and Gates CFS models outperform state of the art unsupervised feature selection algorithms. **(b)**: We fix $k = 20$ and vary the size of each CFS variant's background representation to understand how this hyperparameter impacts feature quality. Our Pretrained and Gates CFS variations continue to outperform baseline methods for all background representation sizes. **(c)**: We vary the relative contribution of grass noise to the dataset and assess the performance of each method when used to select $k = 20$ features.

### 5.1.1 DOWNSTREAM ACCURACY

We began our evaluation by using each of our proposed CFS models to select $k$ features for varying values of $k$ between 10 and 50 with a fixed background representation size $k' = 20$. We evaluated the features selected by each method by training classifiers to predict the digit (i.e., 0-9) in each image. That is, for a given method and value of $k$, we extracted the set of features $S$ selected by that method. The resulting data matrix $X^S$ was then used to train an extremely randomized trees classifier (Geurts et al., 2006), a variant of random forests used by previous works (Drotár et al., 2015; Balın et al., 2019) to evaluate feature selection methods. We emphasize that these classifiers were not used to select features, but only to evaluate the quality of the features selected by each method. We report the accuracy of the classifiers on a held out test set in Figure 4a. We also report results from two other baseline methods that attempt to leverage the weak supervision in the CA setting. That is, we report results from a simple filter baseline that selects the features with the great difference in variances in the target dataset vs. background dataset ("Variances") and for a supervised concrete autoencoder ("Supervised Concrete") trained to distinguish between target vs. background points. We also report results for our fully unsupervised feature selection baseline methods. To reduce visual clutter, we only report results for the best performing fully unsupervised baseline (CAE) in Figure 4a, and provide results for the other baselines in Appendix D.1. We also visualize the features selected by each method in Figure 1 and Appendix D.2.

We find that our Pretrained and Gates CFS models outperform all baseline methods for all values of $k$. On the other hand, we find that our Joint CFS model did not outperform our baseline methods. Such a result potentially indicates that our Joint approach fails to properly disentangle salient and background variations when selecting features.

### 5.1.2 SENSITIVITY ANALYSES

To better understand the robustness of CFS in various settings, we next conducted experiments measuring how CFS's performance was affected by changes in our experimental setup. First, we sought to understand how varying the hyperparameter $k'$ (i.e. the size of the background representation) affected CFS variants' performance. To do so, we fixed the number of selected features $k = 20$ and varied the background representation size between 10 and 50. We then report the accuracy of extremely randomized trees classifiers trained on the features selected from these experiments in

Figure 4b. As a baseline, we also include the concrete autoencoder and our two other weakly supervised baselines trained to select 20 features. For all background representation sizes, we find that our Pretrained and Gates CFS variants continue to outperform fully unsupervised feature selection methods while our Joint variation underperforms baseline models.

We next sought to assess the how the performance of CFS varies with respect to the amount of background noise vs. salient variation. To do so, we varied the scale of the background images of grass used to construct Grassy MNIST. Starting with a scale of 0 (i.e., our target dataset was the standard MNIST dataset), we increased the value of our scale parameter in increments of 0.5 until reaching 4 (i.e., the amplitude of the background was four times that of the digits). For each increment, we used our CFS models as well as our baselines to select $k = 20$ features and evaluated the quality of these features using the same approach described previously. We report our results in Figure 4c. We find that our Pretrained and Gates CFS models consistently outperform baseline approaches for non-zero levels of noise. This result continues to hold even as the level of noise in our data greatly surpasses that of the salient factors of variation. On the other hand, we find that our Joint CFS approach again fails to outperform baseline methods.

### 5.1.3 QUANTIFYING DISENTANGLEMENT

Finally, we sought to better understand why our Pretrained and Gates CFS variants outperformed standard unsupervised methods while our Joint variant failed to do so. In particular, in these experiments we measured how well each of our CFS variants disentangled salient and background variations; that is, we sought to measure how much salient information leaked into each CFS variant's background representations, and how much background information leaked into each variant's selected target features. We expect that methods that struggle to disentangle the two sources of variation would be more likely to select suboptimal features for CA tasks.

To quantify this disentanglement, we reused the CFS models trained to produce Figure 4a. We first extracted the target features selected by each model, and then trained a multilayer perceptron to reconstruct background dataset samples using only these target features. If a method has successfully disentangled background and target variations, thereby selecting a set of target features without much background information, we would expect its selected features to result in higher error on this task. We report the mean squared error of reconstructions for a held out test set in Figure 5a. We find that the features selected by our Pretrained and Gates CFS variants consistently result in higher reconstruction errors than do the features selected by our Joint variant.

Next, instead of extracting the target features selected by each method, we obtained the background representations $g_z(x_i)$ for every $x_i$ in our target dataset. We then trained multilayer perceptrons to reconstruct target data points using these background representations. Better disentanglement again corresponds to higher error on this task. We report our results for this experiment in Figure 5b. We find that the representations learned by our Pretrained and Gates CFS variants result in higher reconstruction error compared to those learned by the Joint model. Taken together, these results indicate that our Gates and Pretrained models are indeed more effective at disentangling salient and background variations than our Joint model. Moreover, when combined with our downstream classification results from Figure 4, these results indicate that achieving such disentanglement is crucial to selecting suitable features for CA tasks.

### 5.2 EVALUATION ON REAL-WORLD DATASETS

We next applied CFS to four real-world datasets. Here we compare our proposed training schemes against the same set of baselines used in our Grassy MNIST experiments. In our evaluations we used each method to select $k = 20$ features for each dataset, except for the Mice Protein dataset for which we used $k = 5$ because of its lower dimensionality. Due to its poor performance on Grassy MNIST, we did not apply the Joint CFS approach to these datasets and only experimented with the Pretrained and Gates variants. We include brief descriptions of our datasets below in Section 5.2.1 along with a table containing the number of samples and features for each dataset in Appendix E.

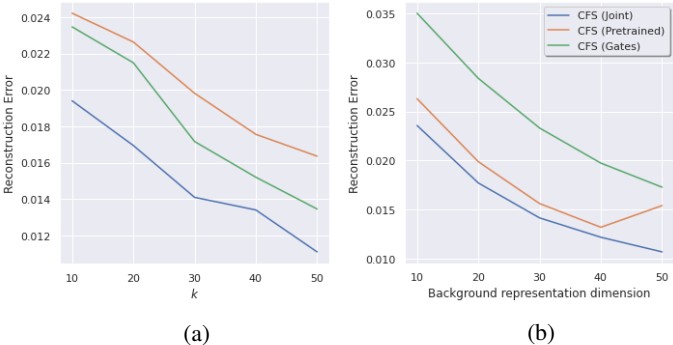

(a)             (b)

Figure 5: **Quantifying disentanglement of background and salient variations.** (**a**): We train multilayer perceptrons to reconstruct *background* samples using the *target* features selected by each CFS variant. A method that successfully disentangles target and background variations should result in higher reconstruction error on this task. (**b**): We train multilayer perceptrons to reconstruct *target* samples using each CFS variant's *background* representations. Once again, higher reconstruction error corresponds to better disentanglement.

### 5.2.1 DATASET DESCRIPTIONS

**Epithelial cell infection response.** We constructed our target dataset by combining two sets of gene expression measurements from Haber et al. (2017). These datasets consist of gene expression measurements of intestinal epithelial cells from mice infected with either *Salmonella* or *Heligmosomoides polygyrus (H. poly)*. Here our goal is to separate cells by infection type. As a background dataset we used measurements collected from healthy cells released by the same authors.

**Bone marrow cell treatment response.** Here we combined datasets from Zheng et al. (2017) containing gene expression measurements from a patient with acute myeloid leukemia (AML) before and receiving a blood cell transplant. Our goal here is to learn a representation that separates pre and post transplant measurements. As a background dataset we used expression measurements from a healthy control patient that were collected as part of the same study.

**Mice Protein Expression.** This dataset (Higuera et al., 2015; Ahmed et al., 2015) consists of protein expression levels from healthy control mice and mice that have developed Down syndrome. Each mouse was injected with memantine, subjected to shock therapy, received both of these treatments, or none of them. Our goal here was to classify mice with Down syndrome based on their treatment regimens. We used data from healthy mice that did not receive any treatment as a background.

**Human Activity Recognition Using Smartphones.** This dataset consists of sensor data collected from smartphones mounted on the waist of study participants as they performed various physical activities (e.g. walking, sitting, laying down), with each data point corresponding to a single activity. The goal of this task was to classify samples by the activity performed during collection. Here our background dataset consisted of measurements taken while participants were laying down.

### 5.2.2 RESULTS

To evaluate each method's performance on these datasets, we once again extracted the features selected by each method and trained extremely randomized trees classifiers to classify target data points. We report the results of these experiments in Table 1. We find that our CFS methods yield consistently higher quality features than do baseline methods, with the sole exception of our pretrained CFS variant on the Mice Protein dataset. We also repeated the same evaluation procedure but with gradient boosted trees as implemented in XGBoost Chen & Guestrin (2016) as our downstream classifier. We report these results in Appendix F and find that they are consistent with our results in Table 1. These results further demonstrate that our Pretrained and Gates CFS variants choose features more suitable for downstream CA tasks than do standard unsupervised feature selection methods.

| Dataset | Lap | MCFS | PFA | CAE | Gated Lap | CFS (Pretrained) | CFS (Gates) |
|---|---|---|---|---|---|---|---|
| Epithelial Cell | 0.593 | 0.606 | 0.596 | 0.834 | 0.638 | **0.904** | 0.897 |
| Leukemia Treatment | 0.708 | 0.853 | 0.588 | 0.934 | 0.644 | **0.970** | 0.952 |
| Mice Protein | 0.865 | 0.820 | 0.838 | 0.793 | 0.784 | 0.847 | **0.973** |
| Smartphone Activity | 0.846 | 0.912 | 0.906 | 0.923 | 0.904 | 0.937 | **0.961** |

Table 1: **Performance of CFS and baseline methods on real-world datasets.** Here we show the classification accuracies of classifiers trained on varying numbers of features selected by CFS and various baseline methods (higher is better). For each method we selected $k = 20$ features (except for the mice protein dataset, for which we use $k = 5$ because the data is lower dimensional). We used an extremely randomized trees classifier and report performance on a held out test set.

## 6 DISCUSSION

In this work we considered the problem of feature selection in the contrastive analysis setting. In this setting we are specifically interested in discovering features that reflect salient variations enriched in some target dataset compared to a background dataset. To tackle this problem we proposed CFS (Section 4). On both simulated and real-world datasets we found (Section 5) that features selected by CFS led to superior performance on downstream CA tasks across multiple data modalities as compared to those selected by fully unsupervised feature selection methods. Like other methods designed for CA, CFS requires that the user have access to a background dataset generated from factors of variation that are uninteresting to the user. Collecting such data may not always be feasible and does impose an additional burden on the user; however, such background datasets are routinely collected in many real-world problem domains (e.g. as part of clinical trials) that are naturally suited for CA. We believe that CFS represents an important first step towards addressing feature selection in the CA setting, and we hope that it can be used as a basis for future work.

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

## A  THE CONCRETE SELECTOR LAYER

The concrete selector layer, introduced in Balın et al. (2019), is based on concrete random variables (Maddison et al., 2016; Jang et al., 2016). A concrete random variable can be sampled to produce a continuous approximation of a one-hot vector. To sample a $d$-dimensional concrete random variable, one first samples a $d$-dimensional vector of i.i.d. samples from a Gumbel distribution (Gumbel, 1954) $\boldsymbol{g}$. Each element $\boldsymbol{m}$ of our concrete variable is then defined as

$$\boldsymbol{m}_i = \frac{\exp((\log \boldsymbol{\alpha}_i + \boldsymbol{g}_i)/T)}{\sum_{k=1}^d \exp((\log \boldsymbol{\alpha}_k + \boldsymbol{g}_k)/T)}$$

Here our temperature parameter $T \in (0, \infty)$ controls the extent to which our one-hot vector is relaxed. In the limit $T \to 0$, our concrete random variable outputs one-hot vectors with $\boldsymbol{m}_i = 1$ with probability $\boldsymbol{\alpha}_i / \sum_{k=1}^d \boldsymbol{\alpha}_k$. Concrete random variables are differentiable with respect to their parameters $\boldsymbol{\alpha}$ via the reparameterization trick (Kingma & Welling, 2013).

We can use concrete random variables to select features through a *concrete selector* layer. The concrete selector layer selects features in the following manner. For each of the $k$ nodes in the selector layer, we sample a $d$-dimensional concrete random variable $\boldsymbol{m}^{(i)}$, $i \in \{1, \ldots, k\}$. The $i$th node in the selector layer $u^{(i)}$ outputs $\boldsymbol{x} \cdot \boldsymbol{m}^{(i)}$. As $T \to 0$ this inner product simply becomes equal to one of the input features. After the network is trained, the concrete selector layer is replaced by an $\arg\max$ layer for which the output of the $i$th neuron is $\boldsymbol{x}_{\arg\max_j \boldsymbol{\alpha}_j}$.

Before training, the selector's parameters $\boldsymbol{\alpha}_j$ are initialized as small positive values to encourage the layer to explore different combinations of input features. As the layer is trained, the values of $\boldsymbol{\alpha}_j$ become more sparse as the layer becomes more confident in particular choices of features.

As proposed in Balın et al. (2019), we train our concrete selector layers using a simple annealing schedule for the temperature parameter $T$. That is, we begin training with a high temperature $T_0$ and gradually decay the temperature until we reach a final temperature $T_B$ at each epoch according to a first-order exponential decay $T(b) = T_0(T_B/T_0)^{b/B}$ where $T(b)$ is the temperature at epoch $b$ and $B$ is the total number of epochs.

## B  DATASET PREPROCESSING

Here we provide preprocessing details for each of the datasets used in Section 5.2. Jupyter notebooks with implementations of our preprocessing workflows can be found in the supplementary materials.

**Grassy MNIST**. We first constructed the original version of Grassy MNIST as described in Abid et al. (2018). For each MNIST digit, a randomly chosen "grass" image from ImageNet was cropped, resized to be 28 by 28 pixels, and scaled to have double (except for the results presented in Figure 4c, for which the scale parameter was varied) the amplitude of the MNIST digit before being super-imposed onto the digit. For a background dataset, we used a separate set of images consisting only of grass. This dataset was used to generate the results in Figure 4a.

To produce the results in Figure 4c, we generated multiple versions of Grassy MNIST using the same procedure described above but with varying the scale of the grass relative to the MNIST digits. In our experiments we varied this scale parameter from zero to four in increments of 0.5.

**Epithelial Cell**. We began by downloading `GSE92332_SalmHelm_UMIcounts.txt` from `https://www.ncbi.nlm.nih.gov/geo/query/acc.cgi?acc=GSE92332`. Each column in this file corresponds to a single cell sample, while each row corresponds to the expression level of a given gene. For each cell, we extracted its condition (i.e., healthy control or disease status) from that cell's label in the first row of the file. In particular, in our experiments we used cells labelled as `Control`, `Salmonella`, and `Hpoly.Day10`; cells with other labels were discarded.

Following standard practices in single cell RNA-seq analyses, we first normalized the counts such that each cell had a total count of 10000 after normalization. We then took the logarithm of these normalized counts + 1. For these two normalization steps we used the `normalize_total` and `log1p` functions implemented in the Scanpy python package Wolf et al. (2018).

**Leukemia Treatment Response**. We downloaded a set of scRNA-seq gene expression measurements from a leukemia patient pre and post-transplant from `https://cf.10xgenomics.com/samples/cell-exp/1.1.0/aml027_pre_transplant/aml027_pre_transplant_filtered_gene_bc_matrices.tar.gz` and `https://cf.10xgenomics.com/samples/cell-exp/1.1.0/aml027_post_transplant/aml027_post_transplant_filtered_gene_bc_matrices.tar.gz`, respectively. A set of measurements from a healthy control patient were also downloaded from `https://cf.10xgenomics.com/samples/cell-exp/1.1.0/frozen_bmmc_healthy_donor1/frozen_bmmc_healthy_donor1_filtered_gene_bc_matrices.tar.gz`.

We then preprocessed our data in a similar manner as done in previous work (Li et al., 2020). That is, we took the log of expression counts + 1, removed any gene features or cell samples consisting entirely of zeros, and filtered down the list of gene features for each file to those that were shared among all the files.

**Mice Protein**. We downloaded `Data_Cortex_Nuclear.xls` from `https://archive.ics.uci.edu/ml/machine-learning-databases/00342/` and converted it to a CSV file. Rows 1 through 77 of this file were used as our features while rows 78 through 81 were used to determine class labels. For a given sample, if a feature value was labeled as missing, we imputed the value of that feature using the mean value for that feature across all samples. As done in previous work (Balın et al., 2019) we then scaled the values of each features to lie between zero and one.

Our background dataset was then chosen to be mice that were healthy, did not undergo shock therapy (label `S/C`) and which received an injection of saline solution. Our target dataset then consisted of mice that did undergo shock therapy (label `C/S`). These target mice were divided into four subclasses based on: whether they were trisomic (i.e., had Down Syndrome) and whether they received an injection of Memantine or a placebo saline solution.

**Smartphone Activity**. We downloaded `train.csv` and `test.csv` from `https://www.kaggle.com/uciml/human-activity-recognition-with-smartphones`. We combined the samples from both files to produce a single dataset (we note that we later performed an 80-20 train-test-split on this combined dataset). For our background dataset, we used samples with the class label `LAYING`, and the remainder of the original dataset was used as our target dataset. Following previous work (Balın et al., 2019) we scaled the values of each feature to lie between zero and one.

## C   IMPLEMENTATION DETAILS

All of our CFS models as well as other methods trained using SGD-based optimization routines (Concrete Autoencoder, Gated Laplacian) were implemented using PyTorch (Paszke et al., 2019) using the Pytorch Lightning[2] API. For all datasets and experiments we used a minibatch size of 128 for methods trained using SGD-based optimization routines.

For our concrete autoencoder implementation, we used two hidden layers of size 512 with ReLU activation functions for all datasets except the mice protein dataset. When training on the mice protein dataset, we used a single hidden layer of size 512 due to the dataset's low dimensionality. At the start of training, we set the temperature of the concrete layer to 10, and gradually annealed the temperature to a final value of 0.1 using the exponential decay schedule described in Appendix A. As in Balın et al. (2019), we trained our concrete autoencoders until the mean of the concrete samples exceeded 0.99.

For all CFS variants, we let our reconstruction function $f_\theta$ be a multilayer perceptron with two hidden layers of size 512 with ReLU activation functions; the only exception was models applied to the mice protein dataset, for which we used only a single layer of size 512. The temperatures of the target feature concrete selector layers were all tuned in the same way as our concrete autoencoder implementation. For our Joint CFS models, we let $g_z$ be a multilayer perceptron with a single hidden layer of size 128. For our Pretrained CFS models, we first trained an autoencoder consisting of an encoder with a single hidden layer of size 128 and a decoder with the same architecture in reverse. The encoder network's weights were then frozen and it was used as $g_z$. Finally, for our Gates

---

[2]`https://github.com/PyTorchLightning/pytorch-lightning`

CFS variant we let $g_z$ be a second concrete selector layer for which the temperature parameter was adjusted as described for our concrete autoencoder implementation. We trained all CFS variants until the mean of the concrete samples exceeded 0.99. For our experiments on Grassy MNIST, unless otherwise specified, we used a background representation size $k' = 20$. For our real-world datasets, we set $k' = 20$ except for the Mice protein dataset for which we set $k' = 5$ due to the lower dimensionality of the dataset.

We trained the concrete autoencoder and gated Laplacian score baselines using the optimizers recommended by their respective authors. That is, for the concrete autoencoder we used ADAM with a learning rate of 0.001, $\beta_1 = 0.9$ and $\beta_2 = 0.999$, while for the gated Laplacian score we used SGD with a learning rate of 1. For all CFS variants we used ADAM with the same hyperparameter settings as for the concrete autoencoder. For the Gated Laplacian, we manually tuned the hyperparameter $\lambda$ so that the method selected the desired number of features for a given experiment.

For the reconstruction experiments presented in Figure 5, we used a multilayer perceptron with two hidden layers of size 512. These MLPs were trained using ADAM with a learning rate of 0.001, $\beta_1 = 0.9$ and $\beta_2 = 0.999$.

For the Laplacian score and MCFS, we used their `scikit-feature`[3] (Li et al., 2018) implementations and ran them with the default hyperparameter choices. For PFA, we used the publicly available implementation from Balın et al. (2019) in the `pfa_selector` function available at `https://github.com/mfbalin/Concrete-Autoencoders/blob/master/experiments/generate_comparison_figures.py`.

To evaluate the downstream classification performance of a subset of features, we trained extremely randomized tree classifiers using the `ExtraTreesClassifier` implementation available in `scikit-learn` with the `n_estimators` parameter set to 100.

To alleviate potential issues with overfitting, we randomly divided each dataset into training and test sets using an 80-20 split. The training data was used to select features and train classifiers for downstream tasks, while the test data was only used to evaluate the performance of the classifiers.

# D  RESULTS FOR ADDITIONAL BASELINES ON GRASSY MNIST

## D.1  QUANTITATIVE RESULTS FOR ADDITIONAL BASELINES

In Figure 6 we provide the results of the Grassy MNIST experiments depicted in Figures 4a and 4c for additional fully unsupervised feature selection methods. We found that for both set of experiments, the concrete autoencoder (CAE) tended to outperform all other baselines tested. No fully unsupervised baselines was able to match the performance of the Pretrained or Gates CFS variants on these experiments.

## D.2  VISUALIZATIONS OF SELECTED FEATURES

In Figure 7 we provide visualizations of the $k = 20$ features selected by each of our CFS variants and fully unsupervised baselines not already depicted in Figure 1. Qualitatively, we observe that the features selected by the Pretrained and Gates CFS variants are of high quality; both methods select a set of features spread out near the center of the image. On the other hand, some of our unsupervised baselines (MCFS, PFA, CAE) and the Joint CFS variant select many features near the edge of the images, where no digit information is present. Moreover, our spectral-information-based methods (Laplacian score, Gated Laplacian) tend to select a small number of clusters of co-located features, limiting the ability of downstream classifiers to distinguish between the ten digit classes.

# E  DATASET DETAILS

In Table 3 we provide additional details (number of samples and features) for each of the real-world datasets used in Section 5.2.

---

[3]`https://github.com/jundongl/scikit-feature`

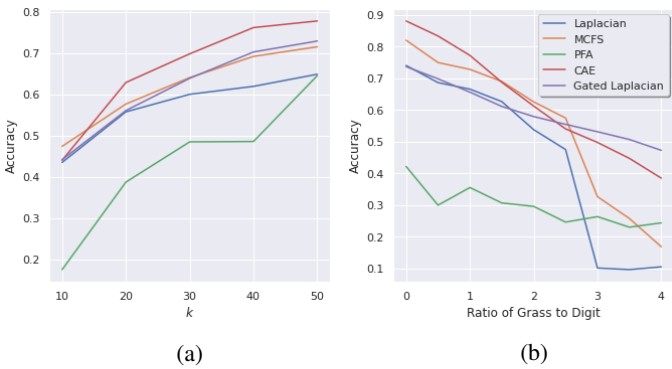

(a)          (b)

Figure 6: **Applying fully unsupervised feature selection methods to Grassy MNIST: (a)**: We measure the quality of features selected by fully unsupervised methods as done in Figure 4a. We find that the concrete autoencoder (CAE) performs the best out of our baseline methods. (**b**): We fix the number of features $k = 20$, vary the ratio of noise to digit when constructing Grassy MNIST, and then evaluate the quality of selected features by each unsupervised baseline as in Figure 4c. Here too we find that CAE tends to perform the best out of our fully unsupervised methods.

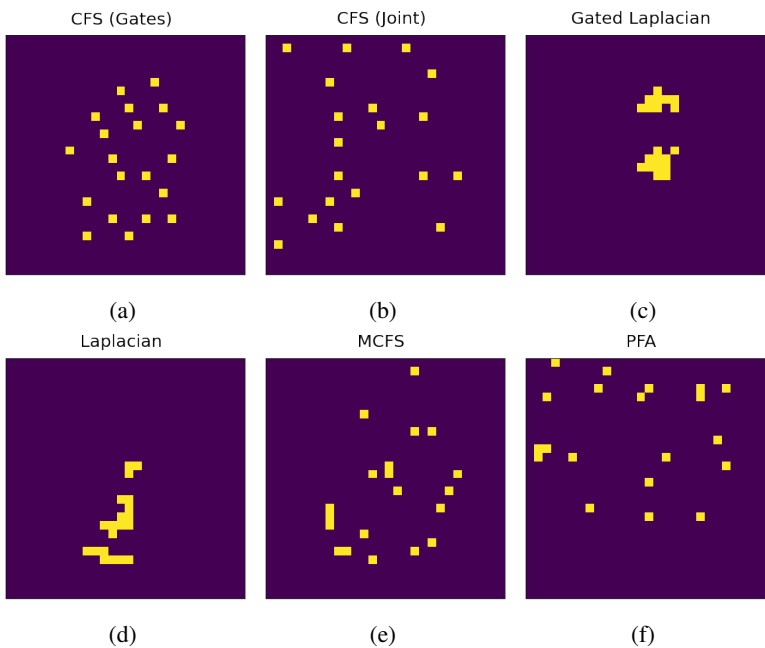

Figure 7: **Additional visualizations of Grassy MNIST results**. Here we provide visualizations of the features selected by each of our baseline unsupervised feature selection methods when run to select $k = 20$ features. We find that our non-spectral-information-based methods tend to pick features that are diffused around the images, with many selected features being close to the edges where digits are never present. On the other hand, our spectral-information-based methods (c, d) tend to select localized clusters of features, which (as demonstrated by our quantitative results), do not capture enough variations to successfully classify images based on digit.

# F   DOWNSTREAM CLASSIFICATION REULTS USING GRADIENT BOOSTED TREES

| Dataset | Features ($d$) | Train size (target) | Train size (background) | Test size | Classes |
|---|---|---|---|---|---|
| Epithelial Cell | 15215 | 3584 | 2592 | 897 | 2 |
| Leukemia Treatment | 12079 | 6318 | 1985 | 1580 | 2 |
| Mice Protein | 77 | 444 | 111 | 108 | 4 |
| Smartphone Activity | 561 | 6684 | 1555 | 1671 | 5 |

Table 2: **Characteristics of real-world datasets used for empirical evaluation.** Each dataset contains both a background dataset, generated from irrelevant sources of variation, and a target dataset, generated from both the irrelevant source of variation as well as the salient source of variation used to generate class labels. We note that CFS uses both target and background data points to select features, while all baseline methods are run solely on the target data points.

| Dataset | Lap | MCFS | PFA | CAE | Gated Lap | CFS (Pretrained) | CFS (Gates) |
|---|---|---|---|---|---|---|---|
| Epithelial Cell | 0.649 | 0.606 | 0.625 | 0.842 | 0.672 | **0.892** | 0.887 |
| Leukemia Treatment | 0.716 | 0.855 | 0.592 | 0.933 | 0.663 | 0.951 | **0.968** |
| Mice Protein | 0.792 | 0.721 | 0.775 | 0.721 | 0.658 | 0.838 | **0.874** |
| Smartphone Activity | 0.829 | 0.876 | 0.906 | 0.908 | 0.873 | 0.936 | **0.940** |

Table 3: **Downstream classification performance of features selected by CFS and baseline methods on real-world datasets assessed using gradient boosted tree classifiers.** Here we show the classification accuracies of classifiers trained on varying numbers of features selected by CFS and various baseline methods (higher is better). For each method we selected $k = 20$ features (except for the mice protein dataset, for which we use $k = 5$ because the data is lower dimensional). Here we used the XGBoost Chen & Guestrin (2016) implementation of a gradient boosted tree classifier and report performance on a held out test set. We find our results to be consistent with those found using extremely randomized trees classifiers (i.e., our CFS variants outperform the fully unsupervised methods).

