# OpenReview forum: "Feature Selection in the Contrastive Analysis Setting"
_ICLR.cc/2022/Conference — ICLR 2022 Submitted_

### Official Review · Reviewer_ft7b · 2021-10-31

**Correctness:** 2
**Technical Novelty And Significance:** 2
**Empirical Novelty And Significance:** 2
**Recommendation:** 5
**Confidence:** 2

**Main Review:**

Strengths. The reported accuracy of the experimental results is extremely good.

Weaknesses. In general, there are a number of unclear points in the paper. Fig. 3: "captures variations due to our uninteresting latent variables z", what do you mean by that? Do you mean that $g_z$ is not useful?

Not clear sentence: "A diverse array of prior work exists on unsupervised feature selection, with one major differentiating factor between methods being how to define the “usefulness” of a feature."

The feature selection is performed using the existing stochastic gating layers, so, the feature selection novelty is absent. The final framework is quite complex, is not extremely clearly described.

In the experimental section, it is mentioned that to assess the quality you train an extremely randomised tree classifiers. Am I right that this part is not drafted on Figure 3? You perform the feature selection (the framework on Fig. 3), and then a classifier (in your case randomised trees) are applied to learn the predictive model?


**Summary Of The Paper:**

Summary. The paper considers the problem of feature selection in the contrastive analysis setting. In particularly, the authors are interested to discover features that reflect salient variations enriched in some target dataset compared to a background dataset. A new method, called Contrastive Feature Selection, is proposed.

**Summary Of The Review:**

The paper presents a combination of an existing feature selection method (called stochastic gating layers) and a classifier (to test the models performance). There is a lack of novelty in the paper.

---

> ### Author Response · Authors · 2021-11-18
> **Response to review**
>
> Dear reviewer,
>
> Thank you for your comments. Based on your “summary of the review”, we believe that the points you listed as weaknesses may be a result of some misunderstandings. As part of our revisions we have worked to improve the clarity of the writing, and we provide detailed responses to your comments below:
>
> First, at a high level our contribution is not to combine existing stochastic gating layers with classifiers as suggested by the reviewer. Rather, we propose a novel architecture that leverages the weak supervision available in the contrastive analysis setting to select features that best represent variations enriched in some target dataset compared to a background. As demonstrated by our experimental results, standard unsupervised feature selection methods perform poorly in this setting, and significant benefits can be achieved by carefully exploiting this weak supervision. Responses to specific comments are below:
>
> 1. **Re: “captures variations due to our uninteresting latent variables z".** The goal of contrastive analysis is to isolate salient variations s present only in a target dataset from the shared variations z also present in a background dataset.  $g_z$ is thus used to isolate information about the shared latent variables z, so that the features selected by our stochastic gates best reflect s (and not z). We have modified this sentence in the newest revision to improve clarity.
>
> 2. **Re: “A diverse array of prior work exists on unsupervised feature selection, with one major differentiating factor between methods being how to define the “usefulness” of a feature”.** Our goal with this sentence was to convey that different methods for unsupervised feature selection use a variety of criteria for selecting features. For example, some methods are based on selecting features with high variance while others (e.g. Laplacian score) select features that best preserve the local structure of nearest neighbor graphs. We have rewritten this sentence in the newest revision to improve clarity.
>
> 3. **Re: novelty.** As pointed out by the reviewer, we do indeed make use of previously proposed stochastic gating layers in our framework. However, as demonstrated by our experiments, when such methods are used naively (e.g. the standard concrete autoencoder), they fail to select high-quality features for contrastive analysis tasks. We thus propose a novel architecture (CFS) that significantly improves the performance of these gating-based feature selection methods in the contrastive analysis setting by carefully exploiting the weak supervision available in this setting. Moreover, as demonstrated by our new experimental results (see revised version of Figure 4), the choice of how to exploit this weak supervision is crucial; more naive approaches do not perform any better than fully unsupervised approaches. We thus believe that CFS represents a sufficiently novel contribution.
>
> 4. **Re: complexity/description of the framework.** We apologize for any confusion resulting from our initial description of the framework. At a high level, CFS attempts to select features that best reflect the variations enriched in a target dataset compared to some background. Our framework thus learns an embedding function $g_z$ to capture variations that are shared across target/background data points. In addition, for target data points we train a feature selection layer to capture features that best reflect variations unique to these target data points. To learn these functions ($g_z$ and the feature selector), they are trained simultaneously with a reconstruction network $f_\theta$ to reconstruct the data points; background data points are reconstructed using only the output of $g_z$, while target data points are reconstructed using both the output of $g_z$ and the feature selector. By only using the selected features to reconstruct target points (and not background points), CFS is able to select features that best reflect variations unique to the target data points. We hope that this description helps clear up some confusion, and have updated the text in the manuscript to be more clear.
>
> 5. **Re: evaluation with extremely randomized trees classifiers.** We apologize for any confusion relating to our descriptions. The reviewer is correct that features are selected using CFS as depicted in Figure 3. The classifiers are not depicted in Figure 3 because they are not part of our proposed feature selection framework; they are simply used to evaluate the quality of the selected features in our experiments. Such an approach is commonly used to assess the performance of unsupervised feature selection algorithms: see e.g. “Concrete Autoencoders for Differentiable Feature Selection and Reconstruction” (https://arxiv.org/abs/1901.09346). We have updated the manuscript to be more clear on this point.
>
>
> Thank you again for your comments.

---

### Official Review · Reviewer_UFq8 · 2021-11-02

**Correctness:** 3
**Technical Novelty And Significance:** 2
**Empirical Novelty And Significance:** 1
**Recommendation:** 5
**Confidence:** 4

**Main Review:**

The article is communicating the proposed approach and conducted exercises in a very clearly and concise manner. Mentioned related work is relevant and credits to prior studies are given appropriately. Both the formulation and solutions proposed appears novel.

Empirical evaluation is performed on augmented MNIST problem (synthetically fused digits with images of grass), where accuracy is measured for different hyperparameter settings, as well as on 4 unaltered biomedical datasets. Proposed approach (in two out of three flavors) showed superior performance in terms of accuracy. Baseline feature selection methods used are exclusively unsupervised, while proposed approach has "unfair advantage" of being trained in a "Contrastive Analysis" setup. That is, proposed Contrastive Feature Selection had a certain insight into downstream problem of classification. It would have been interesting (and highly relevant) to see how would the method compare to the supervised feature selection approaches (e.g using target and background sets as a binary classification problem).



**Summary Of The Paper:**

A new formulation of selecting the top k informative features, when two sets of samples are available: one containing background patterns and patterns of "potential" interest, and the other having background patterns only. Proposed objective is quadratic error minimization between the original example and vector obtained using reconstruction function applied on background embedding vector concatenated with k selected original features. Three different ways of learning the reconstruction and background functions was proposed and evaluated on several datasets (synthetic and real world ones) against unsupervised baselines.

**Summary Of The Review:**

Even though I think the approach can be valuable, my main concern is the motivation to use it when background and target sets are already known. How does its utility compares to supervised feature selection methods? Can Contrastive Feature Selection be applicable to the cases when the target and background samples are not given (or clearly separated)?

---

> ### Author Response · Authors · 2021-11-18
> **Response to review**
>
> Dear reviewer,
>
> Thank you very much for your comments.
>
> To address your concerns re: supervised methods, we added results to Figure 4 from applying two other approaches to Grassy MNIST that make use of the weak supervision (i.e., labels indicating target vs. background) in a more naive fashion. Specifically, we added baseline models that select features by:
>
> 1. Using a supervised feature selection method trained to distinguish between target/background points. We choose to use the supervised version of the concrete autoencoder (a state of the art feature selection method) for this task.
> 2. As suggested by reviewer vsNM, we implemented a simple filter baseline that selects the features which have the greatest difference in variance in the background dataset vs in the target dataset.
>
> For now we have only updated Figure 4a; models are currently being trained so that we can update 4b and 4c as well, but we wanted to share our preliminary results as early as possible during the discussion period. For both baselines, we found that these approaches selected much lower quality features than our gates/pretrained CFS variants based on downstream classification results.
>
> For your concern as to the availability of target vs. background labels: as you have pointed out, CFS is only applicable to cases in which the target vs background distinction is given a priori. However, such labels are naturally collected in a variety of contexts; for example, in data from clinical trials measurements from patients who received a placebo are a natural background for those who received the real treatment. Similarly, when studying a disease, measurements from healthy controls are often taken and are a natural background for those who have the disease. As such, even with this requirement, CFS can be used in a variety of real-world settings.
>
> Thank you again for your comments.

---

### Official Review · Reviewer_vsNM · 2021-11-03

**Correctness:** 4
**Technical Novelty And Significance:** 3
**Empirical Novelty And Significance:** 2
**Recommendation:** 8
**Confidence:** 4

**Main Review:**

Comments:
- The evaluation only uses extremely randomized trees as the classifier, which is not ideal for a feature selection evaluation as the randomness means that it's hard to know if the whole feature space was useful. Some kind of regularized gradient boosted tree, or other non-linear but less randomized classifier would also be helpful in evaluating the selection procedure. Or even an MLP given the feature selection system is in pytorch.
- There are fairly simple filter baselines for contrastive analysis once the problem has been introduced. For example selecting features which have different variances or entropies as compared to the background dataset, or selecting features which maximise the KL divergence between p(x_b) and p(x_t).  Without these baselines it's hard to say whether it's the additional supervision from the background/target difference or the specific techniques proposed in the paper which improve performance.
- Could the authors expand on the difference between the pretrained and gated systems with reference to the experimental results? It's not clear in what circumstances one technique should be preferred over the other, neither experimentally nor theoretically. Presumably the pretrained method took longer as there were two separate training runs, but there are no runtimes reported either.
- Overall the presentation of the contrastive analysis problem is well motivated, the dataset choice and descriptions are helpful and the paper is well written.

Questions:
- How big was the background set for grassy-MNIST?
- Are the grass images sampled with replacement or without replacement, and if without replacement is it the same sample across both background and target datasets? Are any grass images repeated?

**Summary Of The Paper:**

The paper presents a weakly supervised feature selection algorithm which selects features which have different amounts of signal compared across two datasets (a background or control dataset, and the dataset of interest). This allows it to sidestep the requirement for labels to select features for a downstream task or group of tasks. It uses a concrete selection layer to find relevant features with three different approaches for separating them from the background features.

**Summary Of The Review:**

The problem the paper investigates is interesting, the algorithm seems like a sensible extension of unsupervised techniques to the weakly supervised setting.  The experimental study has weaknesses, but still justifies the proposed technique.

---

> ### Author Response · Authors · 2021-11-18
> **Response to review**
>
> Dear reviewer,
>
> Thank you very much for your comments. We are very happy to see that you enjoyed the paper! Responses to your concerns are below:
>
> 1. **Re: using extremely randomized trees.** We originally chose this classifier as it was used to evaluate feature quality by previous works in the literature on unsupervised feature selection (e.g. [1]). To address your concern, we have included additional results using gradient boosted trees (XGBoost) for downstream classification in Appendix F. These results support our original conclusion: i.e., that our pretrained/gates CFS variants outperform fully unsupervised methods in the contrastive analysis setting.
>
> 2. **Re: simpler baselines that use weak supervision.** To address these concerns, we ran two baseline models that have access to the weak supervision on Grassy MNIST, and have added these results to Figure 4. The two baselines are:
>
> - As suggested by the reviewer, we implemented a simple filter baseline that selects the features with the great difference in variances in the target dataset vs. background dataset.
> - As suggested by reviewer UFq8, we trained a supervised feature selection method to distinguish between target vs. background points.
>
> In both cases, we found that these more naive baselines selected lower quality features than our gates/pretrained CFS variants based on downstream classification results.
>
> 3. **Re: background size of Grassy MNIST.** We used an equal number of target and background images during training. Since we used an 80-20 train-test split, this meant that 48,000 background images of grass were used during training.
>
> 4. **Re: grass images used to generate Grassy MNIST.** No grass images were repeated. That is, a different image of grass was combined with each MNIST digit to produce the target dataset, and a separate set of (non-repeating) images of grass were used to generate the background dataset. We refer the reviewer to Appendix B for additional details on how the Grassy MNIST dataset was generated, as well as to “get_grassy_mnist.ipynb” for a Jupyter notebook implementing this procedure.
>
> [1]: Concrete Autoencoders for Differentiable Feature Selection and Reconstruction” (https://arxiv.org/abs/1901.09346)

---

### Official Review · Reviewer_Mgox · 2021-11-04

**Correctness:** 2
**Technical Novelty And Significance:** 1
**Empirical Novelty And Significance:** 1
**Recommendation:** 3
**Confidence:** 4

**Main Review:**

Pros:

1) Modeling the latent vector as salient and background latent vector is interesting.
2) The proposed algorithm does not use too much supervision and only relies on the knowledge of whether a data point belongs to background or target datasets.

Cons:

1) One of the main weaknesses of the proposed method is the choice of k. It is not clear as to how one would choose the number of salient features.

2) I have doubts about the basic problem formulation. The paper talks about contrastive analysis, but completely ignores the recent work on contrastive learning in ML community that uses strong data augmentations and ranking loss functions to generate visual representations that work well for a wide variant of downstream tasks, e.g., Chen et al. 2020 SimCLR.

3) The paper also has some similarity with the notation of isolating factors of variation and the disentanglement literature, e.g., Locatello et al. 2019. In some sense we can think of the underlying problem as identifying the factors of variation in the target dataset that is not present in the background dataset. There is a large literature on disentanglement and representational learning that is relevant to this work.

Francesco Locatello, Stefan Bauer, Mario Lucic, Gunnar Ratsch, Sylvain Gelly, Bernhard Scholkopf, and Olivier Frederic Bachem. Challenging common assumptions in the unsu-pervised learning of disentangled representations. InInternational Conference on MachineLearning, 2019.

4) There is no error bars or statistics for the results reported in Fig. 4. The baseline used is just a concrete autoencoder. Stronger baselines would make the paper strong.

5) The experiment to measure disentanglement is a bit inconclusive. The main idea is that the target features should do poorly in generating the background dataset since it captures both the background and the salient latent variables. The poor reconstruction could be due to many other reasons in addition to capturing salient variables.

6) For grassy MNIST, I am surprised that the authors use a simple MLP architecture. We typically use CNNs and ResNets for imaging data. The use of simple MLP seems like it may not learn much about complex factors of variation. It would be good to provide details on the experimental parameters such as batch size, learning rate, and number of epochs.


**Summary Of The Paper:**

This paper is about feature selection in a weakly supervised setting using a background and target dataset and the goal is to learn features that are specific to the target dataset. The paper formulates the problem using a latent vector for data generation. There is a background latent vector z and a salient vector s. For the background model, the salient vector s is fixed where it has variations specific to the target. The general idea of this paper is to learn an encoding function that takes the input data and generates a low dimensional embedding g_z to captures the variation in z and not the variations in s. We define a loss function that takes this z along with the features to reconstruct the original target points. The main challenge here is that some of the salient features may leak through g_z and we want to avoid this. The paper presents different methods to learn these functions, and the validation is done on a semi-synthetic dataset and a biomedical dataset.

**Summary Of The Review:**

Overall the paper addresses and important problem, but the proposed methods and the experimental results need significant improvement and clarification.

---

> ### Author Response · Authors · 2021-11-18
> **Response to review**
>
> Dear reviewer,
>
> Thank you for your comments. We are very happy to see that you thought our paper addresses an “important problem”. We believe that some of the points you listed as weaknesses may be a result of some misunderstandings. As part of our revisions we have worked to improve the clarity of the writing, and we provide detailed responses to your comments below:
>
> 1. **Re: choice of number of salient features (i.e., k).** The choice for the number of salient features k for CFS is analogous to choosing the number of features to select with a standard unsupervised feature selection algorithm. That is, the user chooses k based on their specific needs. For example, if a user wanted to select 20 genes that best reflect variations of interest in a target gene expression dataset, they would set k = 20. On the other hand, if they wanted 50 genes, they would set k = 50, etc.
>
> 2. **Re: contrastive learning.** Despite sharing “contrastive” in the name, contrastive analysis is entirely unrelated to recent works on contrastive learning for visual/language representations. Works on contrastive analysis [1,2,3] focus on isolating variations enriched in one target dataset as compared to a background, while contrastive learning (e.g. SimCLR) seeks to learn visual representations of images that agree with each other under small transformations. We emphasize that these two lines of work are unrelated, and have added text to the manuscript clarifying this point.
>
> 3. **Re: works on disentanglement.** Thank you for the suggestion. We have added sentences to the related work section discussing the relationship between contrastive analysis and the general problem of learning disentangled representations along with references to work in this area.
>
> 4. **Re: issues with Figure 4.** Thank you for the suggestions. We have added error bars representing the standard deviation over 5 runs to the Figure. As for baselines in the figure: we note that the concrete autoencoder (CAE) is the current state of the art model for unsupervised feature selection (and thus, in our view, is a “strong” baseline). As suggested by reviewers vsNM and UFq8, we have also added results to this figure from baseline models that exploit the weak supervision in the contrastive analysis setting in a more naive way. Specifically
>
> - As suggested by reviewer vsNM, we implemented a simple filter baseline that selects the features with the greatest difference in variances in the target dataset vs. background dataset.
> - As suggested by reviewer UFq8, we trained a supervised feature selection method to distinguish between target vs. background points.
>
> In both cases, we found that these more naive baselines selected lower quality features than our gates/pretrained CFS variants based on downstream classification results.
>
> 5. **Re: disentanglement being inconclusive.** We disagree with the reviewer’s assertion that these results are inconclusive. For example, the only difference between the Pretrained and Joint CFS models is that $g_z$ is trained ahead of time on background points for the pretrained version. Given that this single change is the only difference between these two models, we are not clear as to what other potential causes there are for the differences in reconstruction error aside from the degree of disentanglement.
>
> 6. **Re: choice of architecture for Grassy MNIST.** We chose to use MLP-based architectures for the Grassy MNIST experiments as previous works [2, 3] have shown that simpler transformations (i.e., not using CNNs/ResNets) are sufficient for learning representations that can distinguish between different digit classes for Grassy MNIST.
>
> 7. **Re: providing additional experimental parameters.** These parameters can be found in Appendix C.
>
> [1]: Zou et al., Contrastive learning using spectral methods. In Advances in Neural Information Processing Systems 2013
> [2]: Abid et al., Exploring patterns enriched in a dataset with contrastive principal component analysis. Nature Communications 2018
> [3]: Severson et al., Unsupervised learning with contrastive latent variable models. AAAI 2019

---

### Author Response · Authors · 2021-11-18
**General response to reviews**

Dear reviewers,

Thank you for your detailed and constructive comments. We are thrilled to hear that reviewers found the problem we studied to be “interesting” (reviewer vsNM) and “important” (reviewer Mgox), and we are also happy to hear that reviewers found our method to be “sensible” (reviewer vsNM) and “valuable” (reviewer UFq8).

We have made updates to the manuscript to address your concerns. We have left comments responding to each reviewer’s concerns individually, and we also highlight our responses to some shared concerns below:

1. **Re: stronger baselines that incorporate weak supervision.** As requested by reviewers Mgox, vsNM, and UFq8, we have added results to Figure 4 (the Grassy MNIST results figure) from additional baseline models. These baseline models are based on suggestions from the reviewers on other potential ways to incorporate the weak supervision provided by the target vs background labels; it is worth emphasizing that there are no truly obvious baselines for performing contrastive feature selection because this is a relatively unstudied setting. We find that these baseline models perform no better than fully unsupervised approaches. These results demonstrate that incorporating this weak supervision is non-trivial and further illustrate the significance of our proposed CFS model.

2. **Re: the role of the extremely randomized tree classifier.** We emphasize that this classifier is not part of our proposed framework. The classifier is used only to evaluate the quality of the features selected by each method in the experiments section, as is commonly done in the unsupervised feature selection literature. Following reviewer vsNM’s suggestion, we have also added results from using gradient boosted tree classifiers (XGBoost) instead of ones based on extremely randomized trees.

3. **Re: clarity.** We have made a number of edits to the manuscript to clarify points of confusion raised by the reviewers. For example, we have clarified that our “contrastive analysis” setting is unrelated to the “contrastive learning” performed by methods like SimCLR (note that we are following established terminology). Please see individual comments for more details.

Thank you again for your feedback.

---

### Author Response · Authors · 2021-11-27
**Any remaining concerns?**

Dear reviewers,

Thank you again for your feedback. We would just like to check in to see if our previous responses were satisfactory to address your concerns. If there are any remaining concerns or if anything was unclear in our responses, please let us know.

Thank you

---

> ### Comment · Area_Chair_wAEx · 2021-11-28
> **Re: Any remaining concerns?**
>
> Dear Authors,
>
> Although the reviewers did not directly respond to you, reviewers and I already had multiple-round internal discussion on this paper.
>
> PS: Moreover, I have read all the materials of this paper. You will see the extra comments from me in the meta-review later. Thanks.
>
> Kind Regards,
>
> AC

---

### Decision · Program_Chairs · 2022-01-20

**Decision:**

Reject

**Comment:**

This paper received a majority voting of rejection. During the internal discussion, all reviewers insisted their original scores. I have read all the materials of this paper including manuscript, appendix, comments and response. Based on collected information from all reviewers and my personal judgement, I can make the initial recommendation on this paper, *rejection*. Here are the comments that I summarized, which include my opinion and evidence.

**Research Problem**

In this paper, the authors consider a novel scenario that feature selection in the contrastive setting, where an extra *background* dataset is utilized to remove the background noisy features. However, this problem can be easily handled with a fully supervised feature selection method, where the samples in the *target* datasets are annotated as 1 and the samples in the *background* datasets are annotated as 0. Therefore, the research problem addressed in this paper is not novel. Reviewer UFq8 and ft7b held the same opinion.

**Technical Points**

The technical part could be more informative. The whole framework is based on auto-encoder based self-reconstruction, where the feature selection is finished by the recent CAE model. In my eyes, the major contribution of this paper lie in learning $g_z$, the background representation function. To achieve this, the authors proposed three strategies, *joint*, *pretraining* and *gates*. The *pretraining* idea does not involve any information from the target dataset, where the background representation function is a general one and it has no relationship with the target dataset. I believe the concept of background should be defined based on the target dataset. The *joint* idea suffers from the information leak, which was pointed by the authors. We can also see the inferior performance of the joint model, comparing with two other models. Unfortunately, the philosophy of *gates* is unclear.

**Experimental Evaluation**

(1) The authors only compared with one supervised method on the semi-synthetic dataset. No results of supervised methods on real-world datasets were reported. (2) The performance with different numbers of selected features were not reported.